

# Synthesis of 1,2,3-triazole containing compounds as potential antimicrobial agents

Elise L. Bezold[1], Robert J. Kempton[1], Keith D. Green[2], Ramey W. Hensley[1], Olivia K. Gilliam[1], Sylvie Garneau-Tsodikova[2] and Amber J. Onorato[1]

[1] Department of Chemistry & Biochemistry, Northern Kentucky University, Highland Heights, KY, United States of America
[2] Department of Pharmaceutical Sciences, College of Pharmacy, University of Kentucky, Lexington, KY, United States of America

## ABSTRACT

Invasive fungal infections are increasing worldwide due to an expanding number of immunocompromised patients as well as an increase in drug-resistant fungi. While fungal resistance has increased, this resistance has not been accompanied by the development of new antifungals. A common class of antifungal agents that are prescribed are the azoles, which contain either a triazole or an imidazole group. Unfortunately, current azoles, like fluconazole, have been shown to be less effective with the increase in resistant fungal pathogens. Therefore, the development of novel azole antifungal compounds is of urgent need. The objective of this research was to synthesize triazole-containing small molecules with potent antifungal activity. The scaffold of the synthesized compounds contains a triazole moiety and was synthesized via a copper-catalyzed azide-alkyne click reaction (CuAAC) between the appropriate alkyne and azide intermediates. The minimum inhibitory concentrations of these compounds were determined using standard broth microdilution assays against opportunistic bacteria and fungi associated with life-threatening invasive fungal infections. Although the synthesized compounds possessed no antimicrobial activity, these results can be used to further the long-term goal of developing and optimizing lead compounds with potent *in vitro* antifungal activity.

# INTRODUCTION

Invasive fungal infections (IFI) affect over one million people yearly with a mortality rate that readily exceed 50% worldwide (*Brown et al., 2012*). Therefore, these infections pose an urgent public health threat. Medications for IFIs fall into four categories: polyenes, flucytosine, azoles, and echinocandins (*Perfect, 2017*). Most fungal infections are treated with antifungals from the azole and echinocandins classes; however, systemic fungal infections are treated with Amphotericin B, a last resort antifungal medication in the polyene class (*Perfect, 2017*). The most common pathogens associated with invasive fungal infections are *Cryptococcus, Candida, Aspergillus,* and *Pneumocystis* (*Brown et al., 2012*). Of

Corresponding author
Amber J. Onorato,
onoratoa1@nku.edu

**Figure 1** **Selected antifungals containing 1,2,4-triazoles.**

growing concern is resistance that has been exhibited towards the azole and echinocandin classes of antifungals in *C. albicans*, *Aspergillus*, and *C. auris*, especially in patients with weakened immune systems such as those with HIV, cancer, influenza and COVID-19 (*Fisher et al., 2022*). Therefore, there is significant need to discover novel antifungal agents.

Azoles are the most common clinically prescribed class of antifungals due to their exceptional pharmacokinetic profiles, safety, and minimal drug-drug interactions (*Shafiei et al., 2020*). Many antifungals within this class contain a 1,2,4-triazole, which is exhibited in examples such as fluconazole and voriconazole (Fig. 1) and have broad-spectrum activity against many opportunistic fungal pathogens (*Kathiravan et al., 2012*). In general, triazoles can form non-bonding interactions within the active site of many enzymes and receptors making it an attractive functionality for drug discovery (*Matin et al., 2022*). Azole antifungals work by inhibiting the synthesis of ergosterol, a sterol found in the cell membrane of fungi. Specifically, most azoles exhibit antifungal activity through the inhibition of CYP51 (Erg11), a cytochrome P450 (*Zhang et al., 2019*). As the concern for resistance for the azoles is growing, utilizing an isostere replacement strategy has become commonplace in medicinal chemistry. A well-known isostere of 1,2,4-triazoles are 1,2,3-triazoles which are known to have high chemical stability and solubility in polar biological environments as well as being resistant to metabolic degradation in harsh conditions (*Forezi et al., 2021*; *Singh et al., 2023*). Recently, molecules containing the 1,2,3-triazole scaffold have shown promising activity against severe pathogens such as *C. albicans* (*Perfect, 2017*). Therefore, we sought to investigate the activity of 1,2,3-triazole containing compounds as a substitution for compounds that possess a 1,2,4-triazole.

Armed with this knowledge, we synthesized a series of 1,2,3-triazole compounds (Fig. 2), including several novel molecules, that could possess potency against a series of opportunistic fungal and bacterial pathogens. In our synthesis, the key step was a copper-catalyzed azide-alkyne click reaction (CuAAC) which is an example of an 1,3-Huisgen dipolar cycloaddition (*Rostovtsev et al., 2002*). The use of the click reaction was intentional

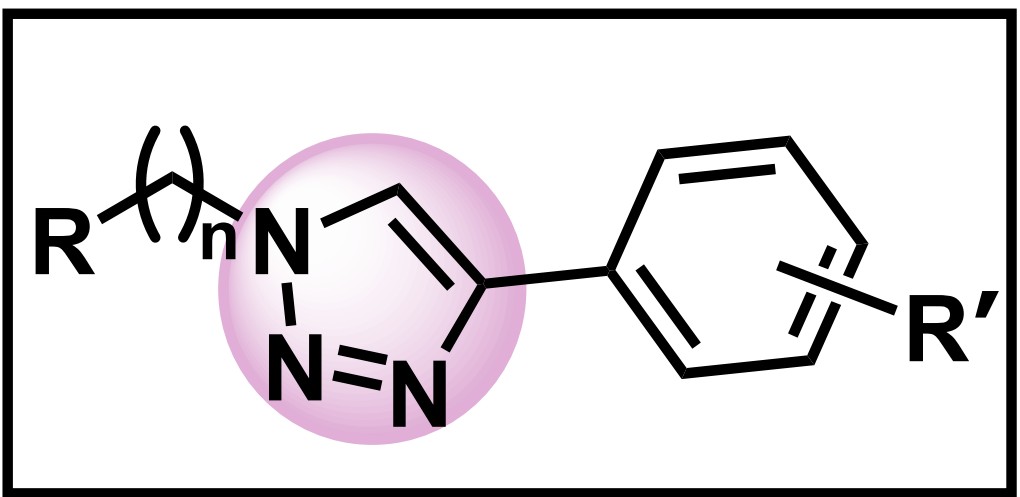

**Figure 2** **Target synthetic molecules.** Our approach was to synthesize compounds with 1,2,3-triazole where $n = 1$ or 2, R = a variety of benzyl and other heterocyclic moieties, and R' = a variety of functional groups as well as various substitution patterns.

as it is a reaction that is high yielding, wide in scope, regiospecific, simple to perform, conducted in aqueous conditions usually at room temperature, and requires relatively simple purification (*Kolb, Finn & Sharpless, 2001*). Herein, we describe the synthesis and biological activity of a library of 1,2,3-triazole compounds.

## MATERIALS & METHODS

All reagents were purchased from commercial sources (Thermo Fisher Scientific, Sigma Aldrich, or Oakwood Chemical) and were used without further purification unless noted. For thin-layer chromatography, Silica XHL TLC Plates, w/UV254, glass backed, 250 um (Sorbent Technologies) were used. Column chromatography was performed using pre-packed RediSep Rf Silica columns on a CombiFlash Rf flash chromatography system (Teledyne Isco). Flash column chromatography was performed on Silica Gel, Premium Grade, 60 Å, 40–63 um (Sorbent Technologies, Norcross, GA, USA). Melting points were obtained using either a Thomas Hoover Capillary Melting Point Apparatus or a Barnstead Electrothermal Mel-temp Capillary Melting Point Apparatus. NMR spectra were obtained using a JEOL 500 MHz spectrometer or a Bruker 500 MHz spectrometer. Most $^1$H NMR spectra were obtained in CDCl$_3$ with CHCl$_3$ ($\delta = 7.26$ for $^1$H) as an internal reference unless otherwise specified. $^{13}$C NMR spectra were proton decoupled and were obtained in CDCl$_3$ with CHCl$_3$ ($\delta = 77.1$ for $^{13}$C) as an internal reference unless otherwise noted. Chemical shifts are reported in ppm ($\delta$). NMR Data are presented in the form: chemical shift (multiplicity, coupling constants (Hz), and integration). The multiplicity patterns are indicated as follows: s, singlet; d, doublet; t, triplet; q, quartet; m, multiplet; bs, broad singlet. High-resolution mass spectra were measured on an VG 70-VSE(B) or a Waters Xevo G2-XS QToF, both with EI+ as the ionization method.

## General procedure for the click reactions

In a round bottom flask charged with a stir bar, the alkyne (1.0 equiv.) was dissolved in THF:H$_2$O (0.08 M, 1:1). In a separate flask, a mixture containing sodium ascorbate (0.50 equiv.), copper (II) sulfate pentahydrate (0.20 equiv.), and THF:H$_2$O (0.08 M, 1:1) was prepared. To this mixture was added the azide (1.0 equiv.) dissolved in THF (approximately 2–3 mL). The azide mixture was then added all at once to the flask containing the alkyne. The resulting mixture was allowed to stir at room temperature overnight. Work up consisted of partitioning the reaction between ethyl acetate and water, washing the organic layer with water, then brine, drying over anhydrous sodium sulfate, and concentration by rotary evaporation. The crude product was typically purified *via* flash column chromatography.

## Bacterial and fungal strains

All synthesized compounds were tested against several Gram-negative and Gram-positive bacterial strains as well as fungal strains to assess their activity, which were obtained from different sources. *Klebsiella pneumoniae* ATCC 27736, *Salmonella enterica* ATCC 14028, and *Escherichia coli* MC1061 were donated by Professor Paul J. Hergenrother (University of Illinois at Urbana-Champaign, Champaign, IL, USA). The bacterial strain *Acinetobacter baumannii* ATCC 19606 and the filamentous fungal strain *Aspergillus terreus* ATCC MYA-3633 were purchased from the American Type Culture Collection (ATCC, Manassas, VA, USA). *Bacillus anthracis* 3452 str. Sterne was provided by Professor Philip C. Hannah (University of Michigan, Ann Arbor, MI, USA). *Enterobacter cloacae* ATCC 13880, *Pseudomonas aeruginosa* ATCC 14028, *Staphylococcus aureus* ATCC 25923, and *Staphylococcus epidermidis* ATCC 12228 were donated from Professor Dev P. Arya (Clemson University, Clemson, SC, USA). *M. smegmatis* mc$^2$122 was provided by Professor Sabine Ehrt (Weill Cornell Medical College, New York, NY, USA). The yeast fungal strain *Candida albicans* ATCC 10231 was acquired from Professor Jon Y. Takemoto (Utah State University, Logan, UT, USA). All bacterial and fungal strains were stored at −80 °C.

## Determination of minimum inhibitory concentration (MIC) values

A total of 10 mg/mL stock solutions of compounds were prepared in DMSO. Fluconazole (FLC), kanamycin A (KAN), and ampicillin (AMP) were purchased from VWR (Radnor, PA, USA) and were dissolved in DMSO (FLC) or ddH$_2$O (KAN and AMP) at final concentrations of 10 mg/mL.

Bacterial strains were streaked on Mueller-Hinton (MH) plates and allowed to grow at 37 °C overnight. 5–10 colonies were scraped from the plate and suspended in MH broth with vortexing to reach an attenuance at 600 nm (OD$_{600}$) of ~0.5. The resulting culture was diluted 1:1000 and added to the prepared serially doubly diluted compound solutions in 96-well plates. For *M. smegmatis*, a 1:100 dilution of the bacteria was done in MH. All MIC values were determined using 96-well plates with low evaporation seals at least in duplicate. For each row, one well was reserved for sterile control (which consisted of 200 µL of MH medium) and another well was reserved for growth control (which consisted of 100 µL of MH medium seeded with 100 µL of bacterial culture).

The MIC values of compounds against yeast cells were determined in 96-well plates as described in the CLSI document M27-A3 with minor modifications (*Clinical Laboratory*

*Standards Institute, 2008b*). A single colony of freshly prepared yeast cells was used to inoculate five mL of yeast extract peptone dextrose (YPD) broth prior to incubation overnight with shaking at 200 rpm at 35 °C. From the actively growing yeast culture, 100 mL were then transferred to 900 mL of RPMI 1640 medium and re-adjusted to achieve $OD_{600}$ of 0.12 ($\sim 1 \times 10^6$ CFU/mL). The cell suspension was further diluted to achieve 1:100 in RPMI 1640 medium. 100 mL of cells were added to the wells of a 96-well microtiter plates that contained compounds prior to incubation for 48 h at 35 °C.

MIC for filamentous fungi were determined as previously described in CLSI document M38A2 (*Clinical and Laboratory Standards Institute, 2008a*). Spores were harvested from sporulating cultures growing on potato dextrose agar (PDA) by filtration through sterile glass wool and enumerated by using a hemocytometer to obtain the desired inoculum size. Serial dilutions of compounds were made in sterile 96-well microplates in RPMI 1,640 medium. Spore suspensions were added to the wells to afford a final concentration of $5 \times 10^5$ CFU/mL. The plates were incubated at 35 °C for 48 h. The MIC values of compounds against filamentous fungi were based on the complete inhibition of growth when compared to the growth control.

## RESULTS AND DISCUSSION

### Synthesis of first generation 1,2,3-triazole compounds

Our primary goal was to synthesize a number of simple compounds that contained a 1,2,3-triazole scaffold in order to determine their potential antimicrobial activity.

The key functional characteristic of our first generation of compounds was the 1,2,3-triazole with a one-carbon linker derived from the corresponding benzyl azide (Fig. 3). In addition, the two aromatic rings possessed a variety of functionality including electron-withdrawing groups (*i.e.,* $CF_3$, F, *etc.*) as well as electron-donating groups (*i.e.,* $OCH_3$, *t*-Bu, *etc.*). The key synthetic step involves a copper-catalyzed azide-alkyne cycloaddition between the necessary azide and alkyne (Fig. 3).

The synthesis of compounds **1–23** (Fig. 3) began with the formation of the azide intermediates. While many were commercially available, we chose to synthesize all azides *via* a substitution reaction between the corresponding benzyl bromide and sodium azide in DMF at room temperature for 3 h (*Sá, Ramos & Fernandes, 2006*). Since all of our synthesized azide intermediates were known compounds, each was verified solely by [1]H NMR and used without further purification.

With the azides in hand, click reactions were performed with a variety of commercially available alkynes to yield compounds **1–23** (Fig. 3). The synthesized 1,2,3-triazole compounds (**1–23**) were obtained in moderate to good yield and provided sufficient quantities for biological testing (see Materials & Methods section for the general procedure).

### Synthesis of second generation 1,2,3-triazole compounds

After the initial set of compounds were obtained, we decided to probe the linker length. With wanting to keep the molecule relatively rigid, we synthesized compounds **24–26** (Fig. 4) from either 1-(2-azidoethyl)naphthalene or 1-(2-azidoethyl)-4-methoxybenzene

**Figure 3** Synthesis of compounds 1–23.

**Figure 4** Synthesis of compounds 24–32.

in order to increase the linker from one carbon to two carbons and followed an analogous synthesis to those previously mentioned.

We also wanted to investigate the use of a benzyl azide derived from a heterocyclic aromatic compound. Based on commercially available benzyl bromides, we chose to synthesize (2-azidomethyl)quinoline, (2-azidomethyl)pyridine, and 3-(2-azidoethyl)-1$H$-indole. As all azide intermediates were known compounds, each was verified solely by $^1$H NMR and used without further purification. With the heterocyclic azides in hand, we

**Figure 5** Synthesis of compounds 33–36.

performed the key click reactions with the corresponding alkynes to yield compounds **27–32** (Fig. 4).

## Synthesis of third generation 1,2,3-triazole compounds

In our final generation of compounds (Fig. 5), we sought to investigate how a carbonyl/hydroxyl functionality would affect the antifungal activity. Our motivation was two-fold. We anticipated that the added functional group would increase the activity; however, if this was not the case, we foresaw further derivatization whereby the carbonyl could be reduced to the hydroxyl.

To begin, 2-bromo-1-(4-fluorophenyl)ethan-1-one was converted to 2-azido-1-(4-fluorophenyl)ethan-1-one using sodium azide as previously described. Following the substitution reaction, the azide was reacted with the alkyne under analogous conditions to provide the carbonyl containing compounds **33–34** (Fig. 5). To further derivatize, compounds **33–34** were reduced using $NaBH_4$ to provide the alcohols **35–36** (Fig. 5).

## Biological evaluation of synthesized 1,2,3-triazole compounds

Once the structure of each compound was confirmed, it was tested against several Gram-negative and Gram-positive bacterial strains as well as fungal strains to assess antimicrobial activity. When assessing the biological activity of our synthesized compounds, we sought to find compounds that possessed single digit $\mu$g/mL antimicrobial activity. To our disappointment, our synthesized compounds did not possess antibiotic or antifungal activity (data tables in supplemental information). Interestingly, compound **14** (Fig. 3) showed minimal activity (16 $\mu$g /mL) toward the bacteria *Mycobacterium smegmatis*. This data was surprising to us given the similarities between compound **14** and inactive compound **13**. Both compounds are derived from the same azide precursor (1-(azidomethyl)naphthalene) but differ in the placement of the fluorine on the aryl group of the alkyne precursor. Compound **14** bears the fluorine in the *ortho* position, whereas

the fluorine substituent is in the *para* position in **13**. We would like to further investigate this result given the structural similarities of these two compounds.

## CONCLUSIONS

In short, a significant number of compounds containing a 1,2,3-triazole moiety were synthesized in moderate to high yields. Though our library of compounds possessed no antibacterial or antifungal activity, the introduction of a number of novel molecules provides other researchers with a relatively easy synthetic route to access these compounds for their own investigations. We are currently working to synthesize our fourth generation of 1,2,3-triazole compounds which will contain two 1,2,3-triazole moieties in hopes of increasing its potency which is inspired by the two 1,2,4-triazole rings exhibited in the antifungal medication fluconazole.

## ACKNOWLEDGEMENTS

We thank the School of Chemical Sciences Mass Spectrometry Laboratory at the University of Illinois at Urbana-Champaign for obtaining HR-MS data.

### Funding

This work was supported by the Department of Chemistry & Biochemistry at NKU, which also obtained HR-MS data; the College of Arts and Sciences at NKU; and Kentucky IDeA Networks of Biomedical Research Excellence (KY INBRE; grant NIGMS 8P20GM103436). This work was also supported by the University of Kentucky. The HRMS data collected at NKU was supported by NSF MRI Award 2116635. The funders had no role in study design, data collection and analysis, decision to publish, or preparation of the manuscript.

### Grant Disclosures

The following grant information was disclosed by the authors:
The Department of Chemistry & Biochemistry at NKU.
The College of Arts and Sciences at NKU.
Kentucky IDeA Networks of Biomedical Research Excellence (KY INBRE): NIGMS 8P20GM103436.
The University of Kentucky.
NSF MRI Award: 2116635.

### Competing Interests

The authors declare there are no competing interests.

### Author Contributions

- Elise L. Bezold conceived and designed the experiments, performed the experiments, analyzed the data, prepared figures and/or tables, authored or reviewed drafts of the article, and approved the final draft.

- Robert J. Kempton conceived and designed the experiments, performed the experiments, analyzed the data, prepared figures and/or tables, authored or reviewed drafts of the article, and approved the final draft.
- Keith D. Green performed the experiments, analyzed the data, prepared figures and/or tables, authored or reviewed drafts of the article, and approved the final draft.
- Ramey W. Hensley performed the experiments, authored or reviewed drafts of the article, and approved the final draft.
- Olivia K. Gilliam performed the experiments, authored or reviewed drafts of the article, and approved the final draft.
- Sylvie Garneau-Tsodikova conceived and designed the experiments, analyzed the data, prepared figures and/or tables, authored or reviewed drafts of the article, and approved the final draft.
- Amber J. Onorato conceived and designed the experiments, analyzed the data, prepared figures and/or tables, authored or reviewed drafts of the article, and approved the final draft.

## Data Availability

The raw data are available in the Supplemental Files.

## Supplemental Information

Supplemental information for this article can be found online at http://dx.doi.org/10.7717/peerj-ochem.10#supplemental-information.

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
