# Peer review of "Synthesis of 1,2,3-triazole containing compounds as potential antimicrobial agents"

_PeerJ Organic Chemistry, doi:10.7717/peerj-ochem.10_

## Round 0.1 · original submission · Major Revisions

Please address the comments of both Reviewers. Most importantly:
* include the original NMR spectra in the SI, even for previously described compounds
* include any missing 13C NMR spectra
* unify the way the NMR spectra are tabulated
* rewrite the main text to remove the reference to SAR, as no true SAR is established in this study
* proofread the text and update/unify the graphics

**Language Note:** The review process has identified that the English language must be improved. PeerJ can provide language editing services - please contact us at [email protected] for pricing (be sure to provide your manuscript number and title). Alternatively, you should make your own arrangements to improve the language quality and provide details in your response letter. – PeerJ Staff

·

Basic reporting

Overall, the article was informative, clearly stated the purpose, and provided excellent methods and results. The references offered sufficient background, the images looked professional. There were several writing errors, while small, would significantly improve the reading experience.
In line 47 of the abstract, in the sentence that begins with “Unfortunately…” sounds like it is discrediting all azoles as antifungal treatments. I believe inserting “current” between “Unfortunately” and “azoles” would help with this issue.

In line 62 – “treatments” needs to be more specific, whether it be drug treatments, therapeutic treatments, or another category. It was surprising to jump from treatments directly into functional group categories. Additionally, is this considered common knowledge or does it need a citation?

In line 63, at the very end of the line it should be “classes” since two classes are named.

In line 64, what is Amphotericin B? Is it a certain class or something else altogether? I would recommend additional clarification on this.

In lines 73 and 74, three antifungals are named but only two are featured in Figure 1. I would suggest adding the third structure, if possible.

In line 79, it has 1,2,3-triazoles end a sentence and the next sentence begins with 1,2,3-triazoles. This is too repetitive and second sentence should be rearranged.

In line 91, I would not mention this makes the project feasible for undergraduates. It is a draw for anyone who wants to consider similar methods that the purification is simple.

Line 112 – round bottom is two words, not one as it appears in the text.

In the SI, compounds 1-4, 13, 14, and 25-36 have the spectral information listed from smallest to largest (ex 5.58ppm, 7.23ppm), however all other compounds report the spectral information from largest to smallest, as it should be. I would also prefer that all new compounds have the image of at least the HNMR data, unless the journal requirements indicate they do not want this information.

Experimental design

The experimental design was clear, could potentially be slightly more concise, and the SI provided any additional information to replicate these experiments.

Validity of the findings

The findings seem valid, with it being unfortunate that the synthesized compounds were not active against the selected pathogens.

Reviewer 2 ·

Basic reporting

Please, see the report given in "Additional comments".

Experimental design

Please, see the report given in "Additional comments".

Validity of the findings

Please, see the report given in "Additional comments".

Additional comments

In this manuscript the synthesis of several 1,2,3-triazole containing compounds is described. The compounds showed no antimicrobial activity, although it is claimed that the results can be used to develop and optimize lead compounds with potent in vitro antifungal activity.
The authors are encouraged to consider the following suggestions:

1. Lines 87-88: Reference 9 is about the CuAAC. Therefore, the authors should refer to the reaction as CuAAC and not as the 1,3-dipolar Huisgen reaction. Of course, the CuAAC is one of the 1,3-dipolar cycloadditions introduced by Huisgen to give five-membered products. In this paper, the authors specifically use CuAAC, so it is better to use that abbreviation to avoid misunderstanding, because there are several 1,3-dipolar Huisgen reactions. Consider also this comment along the text, e.g., lines 181-182.

2. Lines 174-175: It is mentioned the development of a preliminary structure-activity relationship (SAR). However, this has not been carried out in the research described in the paper.

3. Line 182, line 213, line 216: Authors refer to Schemes but those are labeled as Figures. It should be consistent.

4. Lines 190-194: This paragraph is basically a repetition of the general procedure. I believe there is nothing so relevant that cannot be described in the general procedure.

5. Lines 231-232: The authors state that the compounds did not possess antibiotic or antifungal activity at all. So, the paper is reduced to the synthesis of the derivatives. I truly believe that “negative” results (e.g. no bioactivity) should be reported but maybe those should be described in other prestigious journals, which are actually focused on “negative” results with the aim of preserving such useful information. I am not sure if this is the right journal for that.

6. Line 234: The authors state that the data was surprising to them given the similarities between compound 14 and inactive compound 13. However, this discussion should be expanded because it can be very important. What F atom provides is very significant in terms of molecular recognition.

7. Lines 247-250: Obviously, everything must be studied, but the molecular recognition pattern provided by 1,2,4-triazole is not the one given by 1,2,3-triazole. This should be considered in the discussion.

The authors should try to discuss any information about biological targets for the known drugs and potential structural studies that could permit to do a better molecular design.

8. Reference section: The whole reference section should be crosscheck to ensure the same format for each reference.

9. Figures: The subscript numbers R1-R6 should be changed by superscript (they are only labels). Use subscript only to indicate the number of atoms in a given group (e.g. N3)

10. Figures: Please, use the tool “clean-up” in ChemDraw to prevent distortion of the triazole ring.

11. Figures: Copper sulfate is pentahydrate as indicated in the experimental section. However, in the figures it is indicated as monohydrate.

12. Supporting information:

- Copy of 1H-NMR and 13C-NMR should be at least shown to the reviewers (even if the compounds are described in the literature). Ideally, they should be added also to the ESI. It is crucial they show high purity for biological purposes, and the purity is difficult to jugged without seeing the spectra (HRMS is not indicative of purity).
- Ensure that J (coupling constant) is always written in italics.
- Crosscheck the HRMS data. In some cases, the main peak is indicated as M+ but it seems to be MH+
- After compound 4, 13C-NMR data is missing for many compounds that are apparently new. So, the 13C-NMR data should be included.
- For the solid isolated compounds, please provided the melting point.
- Ensure that all chemical structures have same size.
- Some purifications are done using “MPLC”. This is probably a mistake and it refers to HPLC. Please correct it or clarify it.
- Provide Rf values in the case of purifications by column chromatography.
- Compounds 33-36: There is one double bond missing in one of the aromatic rings.

Cite this review as

---

## Round 0.2 · accepted · Accept

The reviewers indicate that their comments have been addressed. The manuscript will be ready for publication once the final error noticed by Reviewer 1 is addressed ("line 63, now on line 67, "class" should be plural since it refers to the two classes stated"). This can be done at the proofing stage.

·

Basic reporting

The article is well-written and the research appears to be sound. The revisions made the article easier to read and added additional clarity. There was only one small writing error that was missed. Previously on line 63, now on line 67, "class" should be plural since it refers to the two classes stated. The inclusion of the NMR data strongly supports the simple purification statement made in the manuscript.

Experimental design

I believe this research fits in well with the scope of the journal. I disagree with Reviewer two's comment 5. The synthesis is significant and the fluoro-containing compound having small anti-fungal activity is interesting and new data. Methods are well done and clear for others to follow.

Validity of the findings

I believe the findings are valid and the data is robust.

Additional comments

I would accept as is, with the minor change to the line 67 for "class"

Reviewer 2 ·

Basic reporting

The paper can be accepted after the revision made by the authors

Experimental design

The paper can be accepted after the revision made by the authors

Validity of the findings

The paper can be accepted after the revision made by the authors

Additional comments

Although I do not completely agree with some opinions stated by the authors, I believe they have donde their best revising the manuscript according to each concern/suggestions. The most critical aspects have been properly addressed and therefore I do recommend its publication.

Cite this review as